# Machine Learning-Based Classification of Cervical Lymph Nodes in HNSCC: A Radiomics Approach with Feature Selection Optimization

**DOI:** 10.3390/cancers17162711

**Published:** 2025-08-20

**Authors:** Sara Naccour, Assaad Moawad, Matthias Santer, Daniel Dejaco, Wolfgang Freysinger

**Affiliations:** 1Department of Otorhinolaryngology-Head and Neck Surgery, Medical University of Innsbruck, 6020 Innsbruck, Austria; matthias.santer@tirol-kliniken.at (M.S.); daniel.dejaco@tirol-kliniken.at (D.D.); wolfgang.freysinger@i-med.ac.at (W.F.); 2Datathings, 5, rue de l’industrie, L-1811 Luxembourg, Luxembourg; assaad.moawad@datathings.com

**Keywords:** head and neck squamous carcinoma, computed tomography, radiomics feature, ensemble method, recursive feature elimination, genetic algorithms, XGBoost, random forest, support vector machine

## Abstract

Head and neck cancers often spread to the cervical lymph nodes, making accurate evaluation vital for treatment planning. Conventional CT criteria based on node size and shape can miss early disease. In this study, we used radiomics, which converts CT images into quantitative features, combined with machine learning to improve classification. Analyzing 234 lymph nodes from 27 patients, our model reliably distinguished between reactive, pathologic, and extracapsular spread nodes. Remarkably, it achieved 90% balanced accuracy using only five key features that align with radiological signs of malignancy. This demonstrates the potential of CT radiomics to provide clinicians with an objective interpretable tool for preoperative lymph node assessment, reducing reliance on invasive procedures.

## 1. Introduction

Head and neck squamous cell carcinoma (HNSCC) is the sixth most common cancer worldwide, with nearly 900,000 new cases and mortality exceeding 50% annually [1,2]. The majority of HNSCC cases are attributed to lifestyle factors, particularly tobacco-derived carcinogens, excessive alcohol consumption, or both. The disease occurs predominantly in the oral cavity, oropharynx, larynx, and hypopharynx—anatomical regions characterized by abundant lymphatic drainage. Consequently, lymph node (LN) status is a critical determinant of diagnosis, staging, and therapeutic strategy [3].

Medical imaging, particularly computed tomography (CT), plays a pivotal role in HNSCC management by providing essential information about tumor characteristics and LN involvement, outperforming physical examination alone [4,5]. The anatomical location of metastatic LN is prognostically significant as dissemination to distinct cervical levels correlates with disease progression and influences therapeutic decision-making [6]. While conventional assessment uses morphological criteria such as axial diameter that exceeds 10 mm, irregular margins, and signs of extracapsular spread [7], these qualitative parameters often fail to reliably differentiate benign from malignant nodes, especially in the early stages. In contrast, CT provides rich quantitative data on the shape, texture, and density of the nodal tissue that could improve diagnostic decision-making.

Radiomics represents a quantitative data-driven approach to medical imaging analysis that extracts and analyzes high-dimensional features such as shape, intensity, and texture from standard imaging modalities [8,9]. By converting medical images into structured mineable data, radiomics enables more reproducible and potentially more objective assessments of tumor and LN characteristics compared to conventional qualitative evaluation. However, radiomics workflows often yield hundreds of candidate features, many of which are highly correlated or redundant. Identifying a compact and diagnostically significant subset that aligns with clinical reasoning remains one of the most technically and conceptually demanding aspects of radiomics. This challenge is further compounded by the small and heterogeneous datasets typical of clinical oncology, which can undermine generalizability and increase the risk of overfitting.

Recent advances in artificial intelligence seek to overcome these limitations by applying radiomics and machine learning to LN evaluation in HNSCC. Deep learning models, particularly three-dimensional convolutional neural networks, have shown promising performance in detecting extranodal extension and metastasis, with area under the receiver operating characteristic curve (AUC) values of 0.84 and 0.90 reported in recent studies [10,11]. However, these models typically require large annotated datasets and operate as black-box systems, limiting interpretability and hindering clinical adoption. In contrast, radiomics models based on predefined hand-crafted features offer greater transparency and alignment with radiological reasoning. However, these approaches often suffer from feature redundancy, high dimensionality, and the need for careful feature selection. Bardosi et al. addressed this by selecting a subset of ten radiomics features and applying linear discriminant analysis, achieving 0.8 accuracy for a three-class LN classification task in HNSCC [12]. While this underscores the potential of interpretable radiomics pipelines, further improvements are needed to enhance performance, robustness, and generalizability.

Our study aims to advance the multi-class classification of cervical LN in HNSCC by distinguishing between three clinically distinct categories: reactive (non-pathologic), pathologic, and pathologic with extracapsular spread (ECS). This classification task introduces additional complexity due to the subtle and often overlapping morphological characteristics shared between these subtypes. To maximize clinical relevance, the proposed methodology employs single-modality CT imaging and interpretable machine learning models, directly addressing the key limitations reported in previous studies.

To address these challenges, we propose an Eliminative Feature Selection (EFS) algorithm designed to identify an optimal subset of features in conjunction with a suitable classifier for accurate LN classification. This approach provides two primary advantages: clinical interpretability, through feature sets aligned with established diagnostic criteria, yet offering quantitative precision beyond visual assessment; and computational efficiency as the EFS process is performed once during training and not repeated at inference. The proposed framework is evaluated using multiple performance metrics—AUC, balanced accuracy (BACC), and F1-score (F1-S)—under a five-fold cross-validation scheme to ensure the robustness and generalizability of the results.

## 2. Materials and Methods

This retrospective cohort study was designed and reported according to the ”Strengthening the Reporting of Observational Studies in Epidemiology” (STROBE) guidelines [13]. Between 2008 and 2020, patients enrolled in the institutional head and neck cancer registry at the Department of Otorhinolaryngology, Head and Neck Surgery, Medical University of Innsbruck, Austria, were screened for eligibility. Inclusion criteria specified patients with newly diagnosed and histologically confirmed advanced-stage HNSCC, classified as stage III or IV according to the Union for International Cancer Control (UICC) staging system. All eligible patients had received definitive primary radiochemotherapy (RCT) as first-line treatment. Exclusion criteria were nasopharyngeal carcinoma, non-HNSCC malignancies, and curative surgery as part of first-line treatment [12].

For each patient, contrast-enhanced CT imaging was available at three clinically relevant time points: baseline staging (staging-CT), treatment planning (planning-CT), and post-treatment response assessment (restaging-CT). This multitemporal approach was employed to capture morphological evolution across the patient care timeline, enhancing the robustness of model development.

From an initial cohort of 1100 potentially eligible patients, 287 met the inclusion criteria, and a random sample of 27 patients (10%) was selected using SPSS version 27 (IBM, Armonk, NY, USA), following modifications based on STARD guidelines [12,14]. In total, 234 LNs were manually segmented in Brainlab Elements (Brainlab, Munich, Germany) and classified by board-certified head-and-neck radiologists with >15 years of CT reporting experience as non-pathologic (n = 81), pathologic (n = 108), or pathologic with ECS (n = 45); see Table 1.

While histopathologic validation of each segmented LN would have required three ultrasound-guided core-needle biopsies per patient and CT scan, significantly increasing the risk of adverse events, classification of nodes was based on established radiologic criteria [15], which are frequently used in clinical practice. More details on imaging, segmentation, and classification are available in Bardosi et al. [12]. CT data were exported in DICOM format, with segmented LNs saved as “Nearly Raw Raster Data” (NRRD) label files.

Ethical Approval and Consent: Locals ethics committee approval has been granted (EK number 1269/2018). The study was conducted according to the guidelines of the Declaration of Helsinki and approved by the Institutional Review Board (Ethics Committee) of the Medical University of Innsbruck (protocol code 1269/2018 and 31 January 2019).

### 2.1. Radiomics Feature Extraction and Data Preprocessing

The radiomics feature extraction process begins by loading CT images along with the corresponding LN masks. Using the PyRadiomics 3.0 library [16], we calculate features from five distinct classes: first-order statistics, shape-based 2D and 3D, and texture class gray-level co-occurrence matrix (GLCM), gray level run length matrix (GLRLM), gray level size zone matrix (GLSZM), gray level dependence matrix (GLDM), and neighboring gray tone difference matrix (NGTDM) features [17]. In total, 120 radiomics features Appendix A are extracted, providing a comprehensive quantification of morphological and intensity-based heterogeneity.

Despite this richness, the dataset presented several analytical challenges. Approximately 2.6% of all feature pairs exceeded the high-correlation threshold of |ρ|>0.9, indicating redundancy within the feature space. With 234 LNs and 120 features—yielding approximately 2.2 samples per feature—the dataset exhibited high dimensionality, increasing the risk of overfitting.

To reduce scale-dependent bias and enhance the comparability of radiomics features, all variables were standardized using Z-score normalization (μ=0, σ=1). This preprocessing step ensured that features with different units or dynamic ranges did not disproportionately influence model training, which is particularly critical for scale-sensitive algorithms like distance-based or parametric classifiers.

Following normalization, the feature distributions were assessed using the Shapiro–Wilk normality test α=0.05. Detailed characteristics of the feature distribution are provided in Appendix A. These results directly informed model selection: normally distributed features (NDFs) were prioritized for parametric classifiers (e.g., linear discriminant analysis and sparse discriminant analysis), ensuring alignment between data characteristics and algorithmic assumptions.

Each LN was represented by three CT scans acquired at different stages of the clinical workflow: staging, planning, and restaging. To ensure data integrity and generalization, unique identifiers were assigned to each LN, enforcing a strict separation between training and test sets at the node level. The dataset was then stratified into training and testing subsets using an 80:20 split while preserving class balance to support unbiased performance evaluation.

### 2.2. Eliminative Feature Selection Algorithms

Eliminative Feature Selection Algorithms (EFSs) play a critical role in radiomics modeling as many pattern recognition techniques struggle with high-dimensional feature spaces and limited sample sizes, which can cause overfitting, higher computational demands, and reduced interpretability [18,19]. By identifying the most reliable and relevant features for model training, EFSs effectively address these challenges.

EFS techniques are broadly categorized into filter, wrapper, and embedded methods, defined by their interaction with the learning algorithm. Filter methods rank features independently using statistical measures (e.g., correlation), prioritizing computational efficiency but ignoring feature interdependencies. Wrapper methods iteratively evaluate feature subsets based on model performance, improving accuracy at the expense of higher computational costs. Embedded methods integrate selection directly into model training (e.g., regularization), achieving a trade-off between efficiency and predictive power [19].

In this study, we employed wrapper-based feature selection methods to identify the most informative radiomics descriptors while preserving their original semantic meaning for clinical interpretability. The goal was to reduce redundancy and dimensionality without compromising classification performance across the three LN classes: “reactive”, “pathologic”, and “pathologic with ECS”. This approach is particularly suited for small datasets as it enhances model robustness by minimizing the influence of correlated features. The following algorithms were used: genetic algorithms, sparse discriminant analysis, recursive feature elimination, as well as random forest and XGBoost, which were also utilized as classifiers.

#### 2.2.1. Genetic Algorithms

Genetic algorithms (GAs) are widely used for feature selection due to their efficiency in exploring large search spaces [20,21]. Inspired by natural selection, GAs evolve a population (a set of candidate solutions) of feature subsets (chromosomes) over successive generations to optimize a given objective function (fitness function). Each subset is encoded as a binary string (chromosome), where individual bits (genes) indicate the inclusion or exclusion of specific features [22].

The process begins with the initialization of a random population of candidate solutions, which are subsequently evaluated using classifiers such as support vector machines (SVMs) or linear discriminant analysis (LDA). Classification accuracy or BACC served as a fitness metric to guide the evolutionary process [23]. High-performing feature subsets are selected through probabilistic strategies, including roulette wheel and tournament selection, prioritizing better-performing subsets to the next generation. To maintain population diversity and prevent premature convergence, crossover operators and mutation operators are applied to generate new feature combinations in successive generations [24].

Multi-objective genetic algorithms (MOGAs) extend traditional GAs by simultaneously optimizing multiple objectives, such as minimizing the number of selected features while maximizing classification performance. Fonseca and Fleming introduced a framework based on Pareto dominance, ensuring that feature subsets are preferred only if they improve accuracy without significantly increasing feature count [25]. A common approach is the weighted-sum method, where the objective function combines feature count and classification performance as follows:(1)$$c=\sum_{i=1}^{2}\lambda_{i}c_{i}\left(\cdot \right)$$
where $c_{1}\left(|P|\right)$ penalizes feature count and $c_{2}\left(P,x,y\right)$ measures classifier performance. The assignment of equal weights ($\lambda_{1}=\lambda_{2}=1$) favors smaller subsets, unless they provide a substantial accuracy gain [26].

A variant known as GAp introduces a soft constraint on the number of selected features, allowing slight deviation from the predefined target size *p* [27]. GA-based feature selection has shown effectiveness in various domains, including bioinformatics and text categorization [19,28]. In particular, Saeys et al. [19] highlighted its relevance in biomedical applications, where dimensionality reduction and interpretability are critical.

Based on these, we implemented a GA framework with extensive optimization settings to ensure reliable convergence and robust feature selection. A population of 5000 individuals was evolved over 500 generations, with fitness evaluated using multiple classifiers: SVM with linear (SVM-L), sigmoid (SVM-S), and radial basis function (SVM-RBF) kernels, as well LDA. To explore the effect of subset size, we evaluated seventeen configurations: from 2 to 10 features (incremented by 1) and from 20 to 90 features (incremented by 10). This tiered approach enabled the detection of both fine-grained and broader radiomics patterns while preserving clinical interpretability.

Due to the stochastic nature of genetic algorithms, we performed 100 independent runs with ensemble averaging to improve result stability and minimize variance. To ensure unbiased evaluation, all subsets of features selected by GA, regardless of the classifier used during selection, were evaluated across every classification model in our study. This rigorous validation framework, which encompassed 27,000 individual experiments, enabled systematic analysis of the balance between feature compactness and predictive power.

#### 2.2.2. Sparse Discriminant Analysis

SDA was introduced by Clemmensen et al. [29] as an extension of LDA by incorporating elastic net regularization, enabling effective feature selection in high-dimensional datasets. Similar to LDA, SDA relies on the assumption of conditional normality: that the dataset X, given a class label *y*, follows a multivariate normal distribution with a class-specific mean vector $\mu_{y}$ and a common covariance matrix ∑, expressed as $X\;|\;y∼N(\mu_{y},\sum )$ [30,31]. This assumption facilitates class separation in theory, but, in high-dimensional settings, unreliable covariance estimation can limit its effectiveness.

To address these challenges, SDA introduces regularization via a penalized optimization framework that enforces both sparsity and solution stability. The objective function is defined as(2)$$\min_{\beta }\left\{{∥\gamma -X\beta ∥}_{2}^{2}+\lambda {∥\beta ∥}_{1}+\gamma^{\prime }{∥\beta ∥}_{2}^{2}\right\}$$
where $X\in R^{n\times p}$ denotes the data matrix, $\gamma \in R^{n}$ encodes class labels, and $\beta \in R^{p}$ represents the discriminant coefficients. The first term minimizes prediction error to improve class separation. The second term, $\lambda {∥\beta ∥}_{1}$, denotes the L1 norm and induces sparsity by promoting feature selection, while $\gamma^{\prime }{∥\beta ∥}_{2}^{2}$, represents the squared L2 norm and enhances stability in high-dimensional settings, particularly when p≫n. Hyperparameters λ and $\gamma^{\prime }$ control the balance between sparsity and regularization strength.

In biomedical imaging, sparsity-driven models such as SDA are particularly valuable in high-dimensional settings, where they offer improved accuracy, computational efficiency, and interpretability, key advantages when dealing with complex and heterogeneous data [32]. SDA functions as a supervised feature reduction method that identifies compact subsets of discriminative features, typically used to train a LDA model. Recent studies have demonstrated the effectiveness of these approaches in tasks that include disease classification, lesion detection, and automated image analysis [33], supporting their integration into a radiomics pipeline.

In this study, SDA was introduced as a dedicated feature selection method within the EFS framework. Prior to this, only NDF features were used without further dimensionality reduction. The SDA-selected subsets were evaluated using six classifiers: random forest, XGBoost, SVM-L, SVM-S, SVM-RBF, and LDA. To assess the effect of subset size on classification performance, 17 configurations were tested, resulting in 90 SDA-based experiments and enabling a comprehensive evaluation of its contribution to the overall pipeline.

#### 2.2.3. Recursive Feature Elimination

Recursive feature elimination (RFE) is a supervised greedy feature selection technique that iteratively removes the least important *n* features, typically n=1, until the target number of features *p* is reached. This method was introduced by Guyon et al. [34] for gene selection in cancer classification; RFE has since been widely adopted in biomedical imaging [35], biomedical signal processing [36], and high-dimensional data analysis, such as DNA microarray studies [37]. By progressively eliminating non-informative features, RFE effectively reduces dimensionality and mitigates overfitting, enhancing predictive accuracy and model interpretability. Extensions of RFE, including integration with *k*-fold cross-validation (RFE-CV), further improve generalizability by stabilizing feature rankings across multiple validation folds [37].

The determination of feature importance in RFE depends on the underlying estimator. In linear models such as SVM or LDA, importance is typically based on the magnitude of the feature weights $\omega_{j}$, whereas, in ensemble methods like random forest or XGBoost, it is derived from criteria such as impurity reduction or permutation importance.

In this study, RFE was implemented with multiple base estimators to enhance feature selection robustness, using the same classifiers as in our earlier EFS analysis. To address variability in stochastic models (random forest and XGBoost), we performed 100 replicates per configuration with ensemble averaging, while deterministic models (SVM and LDA) were run once. Feature selection was performed on 17 predefined subset sizes, consistent with the GA-based setup. All RFE-derived subsets, regardless of the base estimator used, were subsequently evaluated using the full classification pipeline. This experimental design yielded a total of 17,270 individual RFE experiments, enabling a comprehensive assessment of its effectiveness within our framework.

#### 2.2.4. Random Forest

Random forest (RF) is an ensemble learning method introduced by Breiman [38]. It constructs multiple decision trees and determines predictions through majority voting (classification) or mean prediction (regression). The workflow involves bootstrapping, where random subsets of the dataset are sampled with replacement to train individual trees [38]. At each node, a random subset of features is chosen, increasing tree diversity and reducing overfitting [39]. Once trained, predictions from all trees are aggregated, by majority voting for classification or averaging for regression, ensuring model stability and generalization. Additionally, RF provides feature importance scores, making it valuable for identifying key predictors in medical applications such as disease classification and treatment planning [40].

In the medical field, RFs have been widely applied to predict drug demand [40,41], identify longitudinal health trends [42], and classify diseases such as liver disease, breast cancer, and diabetes [40]. In our study, RF was used both as a base estimator within EFS framework and as one of the classification models in the evaluation pipeline. The same experimental strategy applied to other EFS methods was followed: RF-based feature selection was performed across 17 subset sizes, with repeated trials to account for its stochastic nature, and the resulting subsets were evaluated using the full classifier suite to ensure consistency and comparability between methods.

#### 2.2.5. XGBoost

Extreme Gradient Boosting (XGBoost) is an efficient and scalable machine learning algorithm developed by Chen and Guestri [43]. It extends traditional gradient boosting by incorporating second-order Taylor approximation of the loss function, L1 and L2 regularization to prevent overfitting, and parallelized computation for efficient training. These enhancements make XGBoost particularly well-suited for high-dimensional data such as radiomics. The model constructs an ensemble of decision trees iteratively, each tree correcting the residuals of its predecessors. Performance and generalization are optimized by tuning key hyperparameters, such as learning rate, tree depth, and subsampling ratio.

XGBoost has demonstrated excellent results in medical classification tasks. Chang et al. reported 87.5% F1-score in hypertension risk prediction [44], while Inoue et al. achieved an AUC of 86.7% for the prediction of neurological recovery in patients with spinal cord injury [45]. Budholiya et al. further optimized XGBoost with genetic algorithms for heart disease diagnosis, reaching 94.3% accuracy [46]. These findings highlight the versatility and relevance of XGBoost in medical data analysis. In our study, it was utilized both as a base estimator within the EFS framework and as one of the classification models in the evaluation pipeline. Following the same protocol applied to other EFS methods, XGBoost-based feature selection was performed across 17 subset sizes, with repeated runs to address its stochastic behavior. The resulting subsets were then assessed using the complete set of classifiers to ensure consistency and facilitate robust cross-method comparison.

### 2.3. Classification Algorithms

Following feature selection, a diverse set of classifiers were used to model the radiomics feature space and predict LN status. The selection included LDA, SVM, random forest, and XGBoost, which offer a balance of interpretability, flexibility, and robustness to capture varied decision boundaries and accommodate the heterogeneity of radiomics data.

#### 2.3.1. Linear Discriminant Analysis

LDA is a fast and interpretable classification method that improves class separability through dimensionality reduction [30]. It computes the mean vectors of the class, the scatter matrices within the class and between the classes, solving the problem of eigenvalues, and projects the data onto the most discriminative axes, offering a closed-form solution that is efficient for high-dimensional data [47]. However, LDA assumes normality, equal covariance, and linear separability, conditions rarely met in complex datasets [48]. It is also sensitive to outliers and unstable when features outnumber samples. Despite these limitations, LDA remains widely used in radiomics and medical imaging due to its simplicity and efficiency.

#### 2.3.2. Support Vector Machine

SVM is a robust classifier, particularly effective in high-dimensional spaces by leveraging kernel functions to model complex nonlinear relationships [49]. However, its performance is highly dependent on the choice of kernel and hyperparameter tuning, which can be computationally demanding and sensitive to dataset characteristics [50]. Among the commonly used kernels, the linear kernel offers computational efficiency and performs well in linearly separable cases or when the number of features exceeds the number of samples [51]. It is defined as(3)$$K\left(x_{i},x_{j}\right)=x_{i}\cdot x_{j}$$
The sigmoid kernel, inspired by neural network activation functions, enables nonlinear transformations but may suffer from convergence instability and sensitivity to parameter tuning [52]. Its formulation is given by(4)$$K\left(x_{i},x_{j}\right)=\tanh\left(\alpha \cdot x_{i}\cdot x_{j}+c\right)$$
where α and *c* are kernel-specific hyperparameters controlling the shape of the decision boundary.

The radial basis function (RBF) kernel is the most widely utilized, capable of modeling complex decision boundaries by projecting data into an infinite-dimensional feature space [53]. It is expressed as(5)$$K\left(x_{i},x_{j}\right)=\exp(-\gamma ∥x_{i}-x_{j}{∥}^{2})$$
where γ controls the influence of individual training points, with larger values increasing sensitivity to specific samples and smaller values promoting smoother decision boundaries.

Despite the need for careful kernel and parameter selection, SVM remains a powerful and adaptable tool for classification tasks, particularly in high-dimensional small-sample-size contexts relevant to this study.

To consolidate the methodological framework of this study, Table 2 provides an overview of the feature selection strategies and classification algorithms employed, summarizing their underlying assumptions, limitations, and the hyperparameters explored during model development. This synthesis highlights the complementarity of the selected approaches and their alignment with the challenges inherent in high-dimensional small-sample radiomics data.

### 2.4. Cross-Validation Protocol

A stratified 5-fold cross-validation protocol was applied to ensure reliable and unbiased evaluation of model performance. To prevent information leakage, strict LN separation was maintained, with each assigned exclusively to the training or validation set within each fold. The class distributions were carefully preserved in the folds to reflect the composition of the original dataset: 81 reactive, 108 pathologic, and 45 LNs with ECS.

Reproducibility was ensured by applying a fixed random seed (set to 42) during fold partitioning. During each cross-validation iteration, the models were trained on four folds and validated on the remaining fold. Performance metrics (BACC, AUC, and F1-S) were computed for each fold and aggregated across all iterations. Standard deviations were reported alongside the mean values to quantify inter-fold variability and assess the consistency of classification models.

### 2.5. Experimental Framework

The comprehensive evaluation framework (Figure 1) systematically explored the impact of feature selection strategies, subset sizes, and classification models on LN classification performance. Five EFSs were evaluated on seventeen subset sizes and six classifiers, embedded within a rigorous 5-fold cross-validation protocol.

This exhaustive exploration resulted in a total of 44,360 unique experimental configurations. Each configuration was evaluated using three complementary performance metrics: BACC, which quantifies overall classification correctness; AUC, offering a threshold-independent measure of discriminative ability; and F1-S, balancing precision and recall to address class imbalance. By jointly considering these metrics, the evaluation provided a nuanced understanding of the performance of the model across all configurations.

### 2.6. Model Selection Strategy

To determine the optimal classification models for each LN category, an extensive exploration of model configurations was undertaken. This process encompassed a wide range of EFS algorithms, varying subset sizes, and diverse classifier architectures. In total, over 44,360 unique experimental configurations were evaluated, covering the full spectrum of dimensionality reduction techniques and classifiers described in the methodology.

The performance of the model was rigorously evaluated using 5-fold cross-validation via three complementary metrics (BACC, AUC, and F1-S). BACC accounts for class imbalance by averaging sensitivity and specificity, making it well-suited for datasets with uneven class distributions. AUC evaluates the model’s ability to distinguish between classes across all possible thresholds, offering a threshold-independent measure of discriminative power. F1-score reflects the harmonic mean of precision and recall, highlighting the trade-off between false positives and false negatives—both of which are critical in clinical contexts. Together, these metrics captured both threshold-dependent and threshold-independent aspects of performance, providing a robust and clinically meaningful evaluation framework.

We implemented a two-stage selection process to identify optimal models. First, configurations scoring below 0.70 in any metric were excluded to retain only high-performing candidates. Second, a Pareto front analysis was conducted to balance the three performance metrics and the number of selected features. This strategy prioritized models that offered an optimal trade-off between diagnostic accuracy and feature parsimony, providing a robust foundation for subsequent category-specific optimization. The top-performing models of this process were integrated into a class-specific ensemble architecture, allowing customized pipelines for reactive, pathologic, and ECS LN classifications.

### 2.7. Ensemble Method

Ensemble methods are a powerful approach in machine learning that combine the predictive capabilities of multiple classifiers to produce a more robust and accurate overall model [54]. By aggregating the outputs of diverse classifiers, ensemble techniques aim to reduce the variance, bias, and likelihood of overfitting that may be present in individual models. This strategy leverages the principle that a group of weak or moderately strong learners, when combined appropriately, can yield superior predictive performance compared to any single individual constituent model [55].

There are several common approaches to ensemble learning, including bagging, boosting, and stacking. Bagging, or bootstrap aggregating, involves training multiple instances of the same classifier on different subsets of the training data and averaging their outputs to enhance stability and accuracy [55]. Boosting sequentially trains classifiers, with each new model focusing on correcting the errors of its predecessors, thereby producing a strong composite learner. Stacking, which is most relevant to our approach, integrates multiple base models by employing a meta-learner that takes their outputs as inputs and generates final predictions. This framework enables the combination of heterogeneous classifiers, capturing complementary strengths across models [55].

Building upon the optimal configurations identified through our experimental framework, we developed a class-specific ensemble architecture to maximize classification performance across LN categories. For each class, dedicated submodels were constructed, integrating optimized feature selection pipelines, classifier tuning, and probability calibration to improve interpretability and decision confidence. The ensemble employed feature subspace specialization and confidence-based voting, whereby individual model outputs were weighted according to prediction certainty. This tailored strategy enhanced both the accuracy and robustness of the final predictions, providing a reliable decision support system for LN classification.

## 3. Results

Based on the model selection strategy described in Section 2.6, we present the results of a comprehensive evaluation across all the LN classes. A Pareto front analysis was applied to identify the most balanced combinations of EFS algorithms and classifiers, considering both predictive performance (measured by BACC, AUC, and F1-S) and model simplicity, reflected by the number of selected features. For each class, we report Pareto-optimal configurations and highlight the best overall performer in the results tables.

### 3.1. Reactive LN Detection

For the reactive class, four configurations met our selection criteria listed in Table 3:

A detailed analysis showed that the top-performing model combined RFE with an RF estimator to select five key features, with SVM using a sigmoid kernel for classification. The “rfe(RF)_5_SVM-S” combination effectively prioritized relevant features while capturing the moderate nonlinear relationships characteristic of the reactive class. The selected features predominantly included shape descriptors: shape Surface Volume Ratio, shape 2D Diameter Row, shape Maximum 3D Diameter, shape Minor Axis Length, and shape Least Axis Length, reflecting the morphological characteristics of LNs, which were complemented by additional predictive variables identified through the RF importance ranking.

### 3.2. Pathologic LN Detection

For the pathologic class, five configurations met our selection criteria listed in Table 4:

The top-performing configuration combined RFE with an SVM-RBF estimator for feature selection to select seven features, with an RF classifier for final classification. The “RFE(SVM-RBF)_7_RF” selects features emphasizing both textural complexity and node morphology, including glrlm Run Entropy, glrlm Run Length Non-Uniformity Normalized, gldm Dependence Variance, glszm Size Zone Non-Uniformity Normalized, glszm Zone Entropy, shape Maximum 3D Diameter, and shape Sphericity. This hybrid model achieved high AUC and F1-S, demonstrating its effectiveness in capturing pathologic LN.

### 3.3. Pathologic with Extracapsular Spread LN Detection

For the pathologic with ECS approach, nine models met our selection criteria listed in Table 5:

Detailed analysis revealed that the top-performing model combined a GA with an LDA estimator for feature reduction to select five features, with an SVM using a RBF kernel for classification. The “GA(LDA)_2_SVM-RBF” model demonstrated particular sensitivity to textural features glszm Size Zone Uniformity Normalized, complemented by shape Surface Volume Ratio, suggesting that textural irregularities serve as key discriminators for ECS.

### 3.4. Model Comparison and Summary

Table 6 summarizes the performance metrics of the selected configurations across all three classes. While each model demonstrated strengths in its respective class, some exhibited reduced generalization when applied outside their target class.

As detailed in Table 7, the best-performing models relied on distinct subsets of radiomics features tailored to each class. This class-specific selection highlights the heterogeneity of LN characteristics and confirms that no universal feature set can accurately distinguish all classes simultaneously. These findings emphasize the importance of ensemble strategies to leverage complementary information between specialized models and enhance overall classification performance.

### 3.5. Multi-Classifier

To complement class-specific modeling, we implemented a one-versus-rest (OvR) ensemble framework to improve the overall classification across all the LN categories. By decomposing the multi-class task into binary classification problems, the OvR strategy effectively mitigated class imbalance and overlapping feature distributions.

The ensemble method demonstrated strong and consistent performance on all the metrics, as summarized in Table 8. Low standard deviations (Stds.) in BACC (0.030) and AUC (0.028) indicate stable generalization across LN classes, while the slightly higher F1-S deviation (0.085) reflects expected variability in precision–recall balance due to clinical heterogeneity (e.g., reactive vs. ECS-involved nodes). Importantly, this moderate variability underscores the value of class-specific tuning while affirming the ensemble’s ability to generalize effectively.

The AUC curves and confusion matrix (Figure 2) confirm strong class separability with minimal misclassification. Unlike individual class-optimized models, which often show performance drops outside their target class, the OvR ensemble maintained consistently high accuracy across all the categories crucial for clinical use.

## 4. Discussion

HNSCC represents a major global health challenge [1,2,56]. The presence of pathological LNs—with or without ECS—and their anatomical localization are critical prognostic factors, with ECS particularly associated with involvement of laterocervical levels such as level V [3,4,5,6]. The current diagnostic criteria for pathologic cervical LNs on contrast-enhanced CT rely primarily on morphological features, especially shape, which are often subjective and limited in sensitivity [7].

Radiomics provides a powerful tool for extracting quantitative shape, texture, and intensity features from CT images [8,9]. In small datasets common in HNSCC due to rarity and privacy constraints, machine learning remains a viable alternative to deep learning. Recent advances in AI-driven LN classification demonstrate promising results: Kann et al. [10] achieved an AUC of 0.84 for ECS detection using a 3D CNN, while Tomita et al. [11] reported an AUC of 0.90 for metastasis prediction with a deep CNN. Similarly, Bardosi et al. [12] attained 80% accuracy using radiomic features and linear discriminant analysis, underscoring the potential of these integrated approaches.

This study aimed to identify small, compact, and interpretable radiomics feature subsets for each LN class that maintain high classification performance regarding evaluated metrics (BACC, AUC, and F1-S) that are above 0.7. Using a comprehensive grid of 44,360 unique experimental configurations that explore all combinations of feature selection (Figure 1), Pareto front analysis was used to select optimal algorithm combinations, giving equal weight to the three metrics while minimizing the number of features (Table 3, Table 4 and Table 5).

The class-wise performance analysis showed F1-S values of 0.86 for reactive, 0.84 for pathologic, and 0.836 for pathologic with ECS. These results suggest that reactive nodes are more easily distinguishable due to their structural regularity and limited heterogeneity. In contrast, ECS-involved nodes exhibit subtler, more variable features that challenge classification. As shown in Table 7, no single feature set generalizes well across all the classes, and Table 6 confirms that the top-performing algorithm in one class performs poorly in others. This highlights the rationale for employing an ensemble model capable of leveraging class-specific optimizations (Table 8).

The resulting ensemble model achieved macro-averaged scores of 0.889 (F1-S), 0.927 (BACC), and 0.931 (AUC), outperforming prior radiomics-based studies while maintaining model interpretability. These gains are directly attributable to our tailored feature selection, which aligned well with the clinical observations. Reactive nodes were primarily defined by shape-based features, reflecting morphological criteria typically used by radiologists, although such assessments are often subject to interobserver variability. The pathological nodes relied more heavily on texture features, capturing internal heterogeneity that may be difficult to assess visually. The ECS-involved nodes combined first-order and texture features, reflecting underlying asymmetries in intensity and structural disruption that are often under-recognized in clinical imaging. The tailoring of the selection of features and the choice of the classifier to each class improved the diagnostic performance and addressed known sources of variability in radiologic evaluation. These findings support the potential of radiomics as a reproducible and transparent decision-support tool.

The clinical utility of our model is further illustrated by its balanced specificity and sensitivity across classes. Importantly, it minimizes the risk of false negatives—where malignant LNs are misclassified as benign—thereby supporting timely and appropriate treatment decisions. The confusion matrix in Figure 2 reveals limited misclassification between benign and malignant LNs, while the high AUC and recall support the model’s diagnostic robustness.

All 27 patients in this study received primary RCT, and histopathological confirmation of HNSCC was performed on tissue samples from the primary tumor staging panendoscopy [57] without additional ultrasound-guided core-needle biopsies. Four patients diagnosed with carcinoma of unknown primary received an ultrasound-guided core-needle biopsy of the largest suspect cervical LN to confirm the diagnosis of HNSCC. Although this clinical workflow aligns with the NCCN guidelines [58], from a research perspective, a histological or cytological verification of the segmented LNs would have improved the study’s quality. Direct pathological verification of all the segmented LNs, known as the gold standard, would further enhance the research quality, but this would require at least three ultrasound-guided core-needle biopsies per patient and CT scan, an approach associated with increased bleeding risk (1%) and potential tumor seeding (0.001%) [59]. Moreover, core-needle biopsies cannot confirm ECS, which typically requires excisional surgery (e.g., via selective neck dissection). Thus, the radiological definition of “pathologic with ECS” was used, consistent with prior studies [7]. Ultrasound-guided fine-needle aspiration, which carries a lower procedural risk, was also considered [59]. However, fine-needle aspiration is associated with non-diagnostic rates of up to 7% and inconclusive cytology in up to 12% of cases [60], limiting its reliability as a definitive reference. For these reasons, neither ultrasound-guided core-needle biopsies nor fine-needle aspirations were used for comprehensive LN labeling. Lastly, the treatment protocols for HNSCC at our institution are well established and, due to the retrospective nature of this study, could not be altered.

Instead, LN classification—into “pathologic with ECS”, “pathologic”, and “non-pathologic”—was performed by two expert radiologists based on validated CT criteria [15,58], serving as the reference method. While this approach introduces risk of misclassification, it prioritized patient safety and aligned with standard clinical practice.

Despite these promising results, the study is not without limitations. The dataset was relatively small (234 LNs from 27 patients) and derived from a single institution, which may restrict the generalizability of the models. Moreover, the use of radiologist-based CT assessment rather than histopathology as the reference standard for LN classification is relevant. While histopathology remains the gold standard for establishing nodal and extranodal status, preoperative treatment planning in HNSCC typically depends on radiological evaluation. This decision was made to avoid uncertainties in matching preoperative segmented LNs with surgically resected specimens, although it may limit applicability to broader clinical settings. Additionally, although the patient population included variability in LN presentation, broader demographic and imaging diversity is still needed. These limitations underscore the need for future validation using larger multi-institutional datasets and more diverse imaging protocols.

Finally, the explainability of our model remains one of its core strengths. By leveraging compact and interpretable feature subsets tailored to each class, the framework enables transparent and reproducible decision-making. The predominance of shape features in reactive nodes, texture features in pathologic nodes, and a combination of first-order and texture descriptors in ECS-involved nodes aligns closely with established clinical imaging patterns. Preliminary feedback from experienced radiologists confirmed the anatomical and pathological relevance of the selected features, further supporting the model’s clinical validity.

For radiologists and head and neck surgeons, the proposed model presents a practical, interpretable, and reliable tool for pre-treatment lymph node assessment. While not intended to replace histopathological confirmation in surgical cases, the model aims to assist clinicians in making more consistent and objective radiologic classifications—particularly in non-surgical contexts where tissue diagnosis may be infeasible. Accurate identification of ECS-positive nodes holds direct implications for treatment planning, including the extent of neck dissection or radiotherapy field design. By aligning with familiar imaging features and providing a transparent rationale, the model fosters clinician confidence and facilitates integration into everyday diagnostic workflows.

In comparative evaluation with state-of-the-art radiomics classifiers, our approach demonstrated superior accuracy and robustness in distinguishing clinically relevant lymph node classes while preserving full interpretability. This blend of performance and transparency represents a meaningful advancement, facilitating clinical adoption.

Future directions include the integration of deep learning-based segmentation methods, exploration of data augmentation techniques to address class imbalance and enhance model robustness, and prospective validation using larger multi-institutional datasets. To improve generalizability and enable histopathologic correlation, future studies should aim to include HNSCC patients treated with primary surgery. While this may be complicated by difficulties in matching preoperative segmentation with resected lymph nodes, focusing on cases with a single persistent LN following radiochemotherapy may offer a feasible intermediate solution—although such cases are rare and limit statistical power. Additional efforts will focus on incorporating clinical and genomic information for multimodal fusion, as well as benchmarking hybrid models that combine handcrafted radiomics features with deep learning pipelines. Ultimately, our goal is to deploy this compact and interpretable framework as a real-time decision-support tool to support the clinician in non-invasive assessment of LN for HNSCC. For clinical adoption, we propose integrating the model into PACS via a DICOM-compliant plug-in, enabling real-time LN classification directly within the radiology workflow.

## 5. Conclusions

This study introduces an interpretable and class-specific radiomics framework for the multi-class classification of cervical LNs in patients with HNSCC. Through the integration of targeted feature selection and classifier optimization, the model distinguishes between reactive, pathologic, and ECS nodes using a compact set of biologically meaningful radiomics features.

The proposed one-versus-rest ensemble strategy demonstrated consistent performance across classes and reinforced the value of aligning model design with the clinical heterogeneity of LN presentations. By emphasizing interpretability and diagnostic reliability, the framework offers a viable path toward clinical integration.

## Figures and Tables

**Figure 1 cancers-17-02711-f001:**
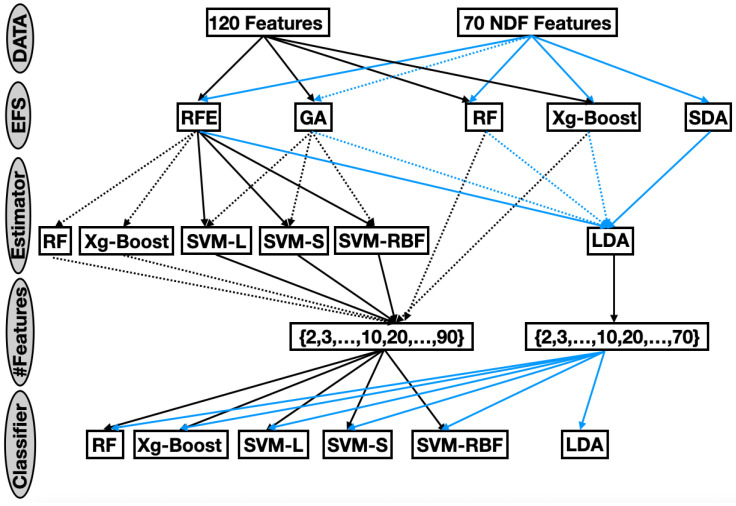
Experimental workflow: Systematic evaluation of 5 EFSs across 17 subset sizes and 6 classifiers. Dashed arrows indicate stochastic components validated through 100 iterations.

**Figure 2 cancers-17-02711-f002:**
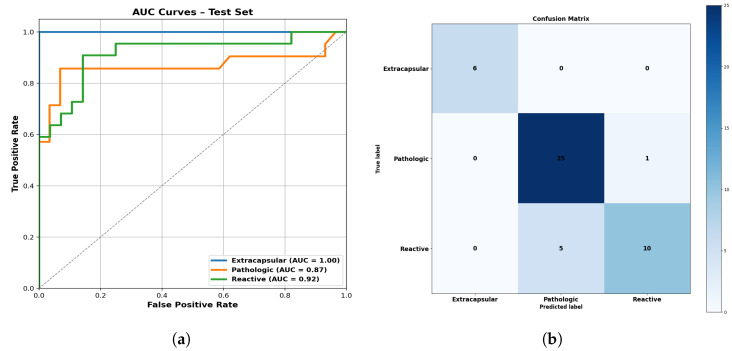
Performance visualizations of the ensemble model. (**a**) AUC for the OvR method across all LN classes. (**b**) Confusion matrix: true vs. predicted labels.

**Table 1 cancers-17-02711-t001:** Summary of patient demographics and distribution of imaging data across treatment phases. Percentages are calculated based on the total number of LNs (n = 234) and patients (p = 27).

Parameter	Group	Number (%)
Age (years)	Mean ± Std.	66.37 ± 9.78
Sex	Male	19 (70.37)%
	Female	8 (29.63)%
Treatment phase imaging	staging-CT	79 (33.76%)
	planning-CT	80 (34.19%)
	restaging-CT	75 (23.05%)
Classes	Reactive	81 (34.62)%
	Pathologic	108 (46.15)%
	Pathologic with ECS	45 (19.23)%

**Table 2 cancers-17-02711-t002:** Summary of feature selection and classification methods. Each method is summarized with its underlying assumptions, known limitations, and selected hyperparameters. All abbreviations used in this table are defined in the Materials and Methods section.

Method	Assumptions	Limitations	Hyperparameters
GA	Population diversity; fitness guides search	High compute cost; sensitive settings; premature convergence	Population = 5000; crossover/mutation rates = 0.5/0.2; generations = 500
SDA	Sparse representation improves generalization	Sensitive to λ, γ; risk of over-sparsity	λ (L1); γ (L2); feature count *p*
RFE	Model ranks features well; reduced sets suffice	Greedy; depends on base model	Feature count *p*; elimination step size; CV folds
RF	Ensemble diversity aids performance	Less interpretable; bias toward multi-level features	Trees count = 200; max depth = 42
XGBoost	Boosting and regularization improve generalization	Sensitive to noise; hyperparameter intensive	Learning rate = 0.01; tree depth; subsample; regularization terms
LDA	Normality; equal covariance; linearity	Outlier sensitivity; unstable in high-dim data	Preprocessing; scaling
SVM	Data separable in kernel space	Kernel/hyperparameter sensitive; costly for large data	Kernel type (L, S, or RBF); γ (RBF); α, *c* (sigmoid)

**Table 3 cancers-17-02711-t003:** Pareto-optimal EFS performances across classifiers for reactive LNs, the highlighted row indicating the best overall performer.

EFS	Estimator	# Features	Classifier	NDF	BACC	AUC	F1-S
GA	LDA	4	SVM-RBF	NDF	0.746	0.829	0.838
RF	–	5	RF	–	0.744	0.713	0.846
RFE	RF	5	SVM-S	–	0.81	0.901	0.86
SDA	–	8	RF	NDF	0.709	0.88	0.768

**Table 4 cancers-17-02711-t004:** Pareto-optimal EFS performances across classifiers for pathologic LNs, the highlighted row indicating the best overall performer.

EFS	Estimator	# Features	Classifier	NDF	BACC	AUC	F1-S
RFE	RF	2	SVM-RBF	–	0.815	0.738	0.735
GA	LDA	4	SVM-L	NDF	0.814	0.748	0.725
GA	LDA	5	SVM-L	NDF	0.805	0.756	0.734
RFE	SVM-RBF	7	RF	NDF	0.805	0.9	0.84
SDA	–	9	RF	NDF	0.761	0.88	0.816

**Table 5 cancers-17-02711-t005:** Pareto-optimal EFS performances across classifiers for pathologic with ECS LNs, the highlighted row indicating the best overall performer.

EFS	Estimator	# Features	Classifier	NDF	BACC	AUC	F1-S
GA	LDA	2	SVM-RBF	NDF	0.856	0.955	0.836
SDA	–	5	RF	NDF	0.889	0.905	0.802
RFE	RF	5	SVM-L	–	0.84	0.965	0.822
XGBoost	–	5	RF	–	0.774	0.976	0.8
RFE	RF	6	RF	–	0.874	0.94	0.822

**Table 6 cancers-17-02711-t006:** Performance metrics for each method and LN class: reactive (green), pathologic (orange), and pathologic with ECS (red). Colored values indicate top performance per target class.

	Reactive				Pathologic				Pathologic with ECS		
	**BACC**	**AUC**	**F1-S**		**BACC**	**AUC**	**F1-S**		**BACC**	**AUC**	**F1-S**
GA(LDA)_2_SVM-RBF	0.706	0.883	0.712		0.805	0.756	0.734		0.856	0.955	0.836
RFE(SVM-RBF)_7_RF	0.681	0.851	0.691		0.805	0.9	0.84		0.674	0.931	0.693
rfe(RF)_5_SVM-S	0.81	0.901	0.86		0.616	0.626	0.587		0.820	0.963	0.787

**Table 7 cancers-17-02711-t007:** Selected radiomics features used for classification of each LN class.

Radiomics Feature	Reactive	Pathologic	ECS
shape Least Axis Length	✓		
shape Maximum 2D Diameter	✓		
shape Maximum 3D Diameter	✓	✓	
shape Minor Axis Length	✓		
shape Sphericity		✓	
shape Surface Volume Ratio	✓		✓
gldm Dependence Variance		✓	
glrlm Run Entropy		✓	
glrlm Run Length Non-Uniformity Normalized		✓	
glszm Zone Entropy		✓	✓
glszm Size Zone Non-Uniformity Normalized		✓	

**Table 8 cancers-17-02711-t008:** OvR ensemble performance across LN classes. Values represent range and Std. across classes.

Metric	Range	Mean	Std.
F1-S	0.815–0.923	0.889	0.085
BACC	0.851–0.952	0.927	0.030
AUC	0.87–0.942	0.931	0.028

## Data Availability

The data presented in this study are available on request from the corresponding author.

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
