# Peer review of "Machine Learning-Based Classification of Cervical Lymph Nodes in HNSCC: A Radiomics Approach with Feature Selection Optimization"

_cancers, 2025, doi:10.3390/cancers17162711_

Round 1

Reviewer 1 Report (Previous Reviewer 1)

Comments and Suggestions for Authors

I have previously reviewed this prior to ammendments.

This version is better. 

For clarity the number of actual patients from which the data is derived should be clear in the abstract.

The explanation for the experienced radiologist report as gold standard for comparison is better explained compared to the previous version, but still remains a flaw in the study; would have been better to study this in patients who had pre-operative CT then surgery so that the real nodal status and extranodal status is confirmed on pathology rather than a radiologist's report.

Author Response

We sincerely thank you for re-evaluating our manuscript and for your constructive feedback.

  1. Number of patients in the abstract – We have revised the abstract to clearly state that the dataset included 234 LNs from 27 patients with HNSCC. This clarification ensures transparency regarding the study population.

  2. Gold standard concern – We fully acknowledge the limitation you highlight regarding the use of experienced radiologist assessment rather than pathological confirmation. This issue has now been addressed more explicitly in the Discussion section:
    "Despite these promising results, the study is not without limitations. The dataset was relatively small (234 LNs from 27 patients) and derived from a single institution, which may restrict the generalizability of the models. Moreover, the use of radiologist-based CT assessment rather than histopathology as the reference standard for LN classification. While histopathology remains the gold standard for establishing nodal and extranodal status, preoperative treatment planning in HNSCC typically depends on radiological evaluation. This decision was made to avoid uncertainties in matching preoperative segmented LNs with surgically resected specimens, though it may limit applicability to broader clinical settings. Additionally, although the patient population included variability in LN presentation, broader demographic and imaging diversity are still needed. These limitations underscore the need for future validation using larger, multi-institutional datasets and more diverse imaging protocols."

We believe these modifications improve the clarity and transparency of the manuscript while honestly acknowledging the methodological limitation.

Reviewer 2 Report (Previous Reviewer 2)

Comments and Suggestions for Authors

hello

thank you for this improved paper

there are some major improvements when comparing old and this new version

in current paper, the topic is sound, it's explained and highlights some new and future directions in oncology

its worth to notice that all important facts are now more visible and clear

used figures and tables are OK

abstract is sound and corresponds with the improved paper in total

Authors did wrote the paper in a very high standard with improved overal meaning

current format and structure of the paper is very good

I dont see any major mistakes here, the paper should be published

when comparing the changes from old and new paper, I like the new version a lot

I hope this paper will be a source for many future citations 

With kind regards

Author Response

I would like to sincerely thank you for the time and effort you dedicated to reviewing my paper. Your thoughtful evaluation and constructive perspective have been invaluable in helping improve the manuscript to its current form.

I am deeply grateful for your recognition of the improvements in clarity, structure, and presentation, as well as your encouraging words about the overall quality of the work. Your positive feedback about the abstract, figures, and tables, as well as your view that the paper now reaches a high standard, means a great deal to me.

It is truly rewarding to know that the revised version resonated with you.

With kind regards and sincere appreciation,

This manuscript is a resubmission of an earlier submission. The following is a list of the peer review reports and author responses from that submission.

Round 1

Reviewer 1 Report

Comments and Suggestions for Authors

The authors have made presumption that readers will know what certain technical terms mean eg. AUC without explaining its relevance.

The standard comparison for the AI is the radiologist report. However radiologist report is not the gold standard, as they can often get the reactive, pathological and extracapsular spread wrong. The gold standard for comparing reactive vs pathological (in a patient not having surgical treatment) is a fine needle aspirate of the node not the radiologist reports.

The gold standard for extracapsular spread is pathological examination of the nodes which is not done here, radiologist report has not been shown to be accurate (from literature radiology extranodal spread misses 28-40% of those that were proven on neck dissection).

Currently the utility of AI predicting ENE have little implications for radiotherapy treatment, hence I feel there is limited practical usefulness of this study, but it is an interesting concept that require further development. 

I think the study would have been more useful if it were used in patients who will be undergoing surgery (neck dissection) where one can compare the AI prediction of ENE with the actual pathological response. Therefore I feel that the study methology is flawed when using radiologists report as a gold standard.

Author Response

Dear Editors,

We sincerely thank the reviewer for this valuable and constructive feedback, which helped us improve both the clarity and scope of our manuscript.

Regarding the use of technical terms such as AUC, we ensured that all abbreviations are now defined either in-text or in table legends and explain the clinical relevance.

We also fully acknowledge the reviewer’s critical point concerning the use of radiologic assessment rather than histopathology as the reference standard. This study is retrospectively analysing patients that were treated according to our clinical protocols and oncologic guidelines. The ethics' committee approval is valid for retrospective analysis only. Furthermore quality assured manual lymph node segmentations are the basis of this investigation and the therapeutic route. These segmentations, however, are standard of practice in our radiologic diagnosis and radiotherapy routines and follow established quality assurance policies. The approach presented in this paper is thus closely aligned with clinical reality in our hospital. In a currently ongoing work we evaluate the performance of deep learning based lymph node segmentation and classification in combination with the present approach.

While we recognize the limitations of radiologic labelling, the reference classifications used in our dataset follow validated imaging criteria and reflect real-world decision-making in non-surgical management of HNSCC. We have clarified in the revised Discussion and Limitations sections that this approach does not represent a definitive gold standard but rather a clinically grounded approximation. We also added a reference detailing the criteria used by board-certified radiologists to support reproducibility.

As for the clinical implications, The high classification of the approaches presented in this work would allow a quick and simple classification of lymph nodes already at the beginning of radio-chemotherapy and could serve as an additional tool to furthers establish ML based support of oncologic treatment for HNSCC. This has now been emphasized in the revised Discussion section.

Finally, while we agree that validation in a surgically treated cohort would strengthen the findings, our intentional focus on non-surgical patients addresses a common and underrepresented clinical setting in which imaging remains the only available modality for lymph node characterization. We have clarified this rationale and more explicitly positioned our study as a proof-of-concept that supports future research using histopathologic ground truth.

Once again, we are grateful for the reviewer’s comments, which helped us improve methodological transparency and define the scope and clinical relevance of our work more clearly.

Sincerely,
Sara Naccour
PhD Student: Medical University of Innsbruck, Austria
sara.naccour@i-med.ac.at

Reviewer 2 Report

Comments and Suggestions for Authors

hello

thank you for this interesting paper

the abstract - is missing, along with key words and title

please edit this important section

without abstract the paper is worthless - look the pdf file that downloads itself

title/key words - its hard to validate - not included in the main paper pdf.

  1. introduction - the first page is missing or deleted? furthermore, looks like this pdf format is not suitable for any mdpi journals, probably authors copied data and forget to format the paper as well. the introduction is quite long but its very important. in the introduction nothing has to be changed, a clear aim is missing - its a lot of words and sentences missing one key aim. At the end of paragraph 1 authors wrote many possible aims and targets but which one is it? Introduction is long and very discussable, perhaps move some parts to discussion
  2. material and methods - this chapter is very interesting. authors wrote clear inclusion criteria and did decsribe each methodological step in a great detail with all data present. this chapter is very well presented, step by step and with good detail. so far I dont see any necessity to improve any of this chapter structure. the methodology itself is well build and decsribed on selected participants, that meet the inclusion criteria for the study. some valuable insights and informations are presented in chronological points that make this chapter easy to read and understan - nothing to change here
  3. results - what is this green line in table 3? why   it is not described in text, and its meaning? some figure/table legends abbreviations needs some improvement in decsription, so that they will be more suitable to understand for readers. The introduction to results are somehow hard to understand - explain in please. results are presented in sub-chapter according to their meaning. the colored values in table 6 needs explanation in test and table abbreviations - I dont know what do they represent. Somehow, while reading this paper I hace this feeling that it was copied from another manuscript but not with great care to detail? - Results are organised and described in detail. All mathing figures and tables are sufficient, nothing to change
  4. discussion - is quite short and doesnt discuss all authors results, which are many in this paper. Used references needs improvement. Authors did not present any study limitations and did not highlight top key important clinical remarks from this study. I dont feel the discussion suitable in this paper - that needs improvements. What exactly are the benefits for a normal working surgeon from this study? Does used CT have some limitations? why not all more than thousand participants were enlisted, and what makes them so special? - furthermore why this special inclusion criteria for the study is not discussed in the discussion section? its good that the authors presented some possible future perspectives. how this paper and the presented results might improve each clinician work in regard to HNSCC?
  5. conclusions are OK, references needs improvements. at the end of the study there is no information about nay bioethics comitee approval as well as all included patients in the study signed a form to participate in the study - or perhaps if its really a retrospective study, what necessary legal forms were obtained?
  6. references needs improvement - poition 45 is from 1936 - is it really needed? is this paper for clinicians or for historicians?

Author Response

Dear Editors,

We sincerely thank the reviewer for their thorough and thoughtful evaluation of our manuscript. We appreciate the constructive feedback, which has helped us significantly improve the quality, clarity, and clinical relevance of the work. We have carefully addressed each of the concerns point by point below:

  1. We regret the oversight and thank the reviewer for pointing this out. The missing abstract, title, and keywords were due to a formatting issue during PDF generation and submission. We have corrected the file structure and ensured that the title, abstract, and keywords now appear properly in the revised submission in accordance with MDPI guidelines.
    We have now carefully reformatted the manuscript using the official MDPI LaTeX template, ensuring all front matter, pagination, and formatting conform to MDPI requirements.
    The introduction was revised to improve clarity, structure, and alignment with the study’s objectives. The content was reorganized to follow a logical progression from the clinical background of HNSCC to current imaging limitations, and finally to the rationale for our proposed method. The central aim—developing an interpretable, CT-based machine learning framework for multiclass lymph node classification—was stated more explicitly. Redundant or overly detailed content was removed, making the introduction shorter while preserving all essential information. The tone and terminology were also refined to enhance readability and meet academic standards.
  2. Thank you.
  3. We revised the opening paragraph of the Results section to clarify the structure and logic of the analysis. We now explicitly explain how the experiment was organized and how the results are grouped and interpreted.We have updated the legends and table captions to explain:
  • The green highlight in Tables 3, 4, 5, indicates the top-performing configuration based on Pareto front analysis.
  • The color-coded values in Table 6, reflect the best metric scores per class or classifier. These are now described in the table captions and mentioned in the text for clarity.
  1. The Discussion section has been significantly revised to:
  • Interpret key results in more clinical and practical terms.
  • Discuss study limitations, including the retrospective design, the manual nature of LN segmentation, and limitations of using CT alone.
  • Justify the sample size: We now clarify that from the initial cohort of 1,100+ patients, only those with complete, tri-phasic CT imaging and no prior surgery were included to ensure consistency and modeling integrity. We were using in the patients cohort of paper “..” Where  all pertinent details are exposed.
  • Highlight clinical implications, including how interpretable machine learning models can support radiologists and surgeons in distinguishing lymph node classes and potentially reduce unnecessary interventions. We state in the discussion that only a few Radiomics features suffice to reach very high “diagnostic” accuracy.
  1. We have now added a dedicated Ethics Statement at the end of the manuscript.
  2. We appreciate the reviewer’s attention to the reference list. The cited work (Fisher, 1936) refers to the original formulation of Linear Discriminant Analysis (LDA), which serves as the theoretical foundation for one of the classifiers used in our study. 

We appreciate the reviewer’s time and insightful comments, which have led to substantial improvements in the clarity, structure, and clinical relevance of the manuscript. We believe the revised version now addresses all points raised and is fully aligned with MDPI submission standards.

Sincerely,
Sara Naccour
PhD Student: Medical University of Innsbruck, Austria
sara.naccour@i-med.ac.at
